# Onabotulinumtoxin A for the Treatment of Post-Traumatic Headache: Is It Better than Anti-CGRP Antibodies?

**DOI:** 10.3390/toxins16100427

**Published:** 2024-10-02

**Authors:** Lanfranco Pellesi, Dilara Onan, Paolo Martelletti

**Affiliations:** 1Clinical Pharmacology, Pharmacy and Environmental Medicine, Department of Public Health, University of Southern Denmark, 5230 Odense, Denmark; 2Department of Physiotherapy and Rehabilitation, Faculty of Health Sciences, Yozgat Bozok University, Yozgat 66800, Türkiye; dilaraonan@gmail.com; 3School of Health, Unitelma Sapienza University of Rome, 00161 Rome, Italy

**Keywords:** migraine, pain, prophylaxis, PACAP, traumatic brain injury

## Abstract

Post-traumatic headache (PTH) is a common and debilitating consequence of traumatic brain injury (TBI), often resembling migraine and tension-type headaches. Despite its prevalence, the optimal treatment for PTH remains unclear, with current strategies largely extrapolated from other headache disorders. This review evaluates the use of onabotulinumtoxin A (ONA) and anti-calcitonin gene-related peptide (CGRP) monoclonal antibodies (mAbs) in the treatment of PTH. A comprehensive literature search was conducted on PubMed, including studies published up to September 2024, focusing on the efficacy, safety, and mechanisms of onabotulinumtoxin A and anti-CGRP mAbs in PTH. Both clinical trials and observational studies were reviewed. ONA, widely recognized for its efficacy in chronic migraine, has shown limited benefits in PTH with only one trial involving abobotulinumtoxin A in a cohort of 40 subjects. A phase 2 trial with fremanezumab, an anti-CGRP monoclonal antibody, failed to demonstrate significant efficacy in PTH, raising questions about the utility of targeting CGRP in this condition. ONA may offer advantages over anti-CGRP mAbs, not only in terms of its broader mechanism of action but also in cost-effectiveness and higher patient adherence. Both ONA and anti-CGRP mAbs are potential options for the management of PTH, but the current evidence is insufficient to establish clear guidelines. The negative results from the fremanezumab trial suggest that CGRP inhibition may not be sufficient for treating PTH, whereas onabotulinumtoxin A’s ability to target multiple pain pathways may make it a more promising candidate.

## 1. Introduction

Post-traumatic headache (PTH) is classified as a secondary headache in the International Classification of Headache Disorders (ICHD), emerging within seven days following trauma, regaining consciousness, or the ability to perceive pain [1]. Diagnosis of PTH is based on specific criteria, with persistence beyond three months indicating chronicity. The clinical presentation often overlaps with migraine and tension-type headaches, necessitating comprehensive neurological assessments, including soft tissue palpation and range of motion measurements, to refine the diagnosis. Migraine is a recognized risk factor for developing PTH, with both conditions sharing symptoms such as photophobia, phonophobia, nausea, and vomiting [2]. The temporal association of headache onset with a causative injury is a key diagnostic criterion for PTH [3]. Each year, approximately 69 million individuals worldwide suffer from traumatic brain injury (TBI), with headache being the most common post-TBI complaint, affecting 30% to 90% of patients [4]. Notably, 18% to 22% of individuals continue to experience headache one year post-injury [5,6]. PTH is twice as prevalent in women compared to men, with risk factors including a history of migraine, female gender, young age, severe head injury, and comorbid psychological conditions [7,8]. The pathophysiology of PTH remains incompletely understood, though several mechanisms have been proposed, including impaired descending pain modulation, neurometabolic alterations, and activation of the trigeminal sensory system [3]. Post-TBI, both structural and functional remodeling occur in cortical and subcortical brain regions, disrupting normal pain modulation and potentially leading to PTH. Neurometabolic changes following TBI can result in oxidative stress, ion homeostasis disruption, and cortical spreading depression, all of which may contribute to the development of PTH [9,10,11]. Additionally, the role of calcitonin gene-related peptide (CGRP) in PTH pathogenesis has been suggested [12,13].Current treatment strategies for PTH largely mirror those used for other headache disorders, with a multimodal approach often recommended due to the limited number of randomized controlled trials with specific treatment protocols [14]. Acute management may involve triptans, nonsteroidal anti-inflammatory drugs (NSAIDs), and antiemetics, while prophylactic options include anticonvulsants and tricyclic antidepressants [15]. For migraine-like PTH, treatments such as onabotulinumtoxin A (ONA) and anti-CGRP monoclonal antibodies (mAbs) have gained attention. Both injectable treatments are of particular interest for PTH due to the overlapping features with migraine, though clinical studies in PTH remain limited. This review aims to summarize the current evidence on the use of ONA and anti-CGRP mAbs in the treatment of PTH, with additional considerations regarding the potential advantages of one treatment modality over the other.

## 2. Onabotulinumtoxin A

ONA is a well-recognized treatment for chronic migraine [16]. ONA inhibits the release of acetylcholine at the neuromuscular junction, leading to a blockade of neural stimulation and resulting in temporary muscle paralysis depending on the dosage [17,18]. Its analgesic effects are attributed to the inhibition of pain mediators, including CGRP, substance P, and glutamate. Specifically, ONA cleaves SNAP25, a protein involved in the release of neurotransmitters, reducing the exocytosis of pain mediators like CGRP and substance P from sensory neurons [19,20]. This inhibition indirectly reduces the activation of transient receptor potential vanilloid 1 (TRPV1) receptors, which are involved in pain perception, leading to reduced peripheral sensitization and neurogenic inflammation [21]. In an animal model of PTH following mild TBI, ONA administered shortly after injury prevented both acute and stress-induced allodynia for up to 14 days, suggesting that early intervention with ONA could prevent the progression from acute to chronic PTH [22]. A case series showed the effects of ONA in three patients with PTH following gunshot wounds to the head, neck, and face [23]. Despite previous treatments for head and neck pain, cervical dystonia, and muscle spasms, the patients experienced substantial improvements in pain intensity, headache frequency and duration, and muscle spasms after ONA administration. For instance, in one case, a patient with a visual analogue scale (VAS) score of 8 for headache intensity reported a reduction to 1 after one month, with sustained improvement after three months. Other cases similarly demonstrated reductions in headache intensity and frequency, with long-term benefits reported up to nine months post-treatment. A case report involving a 62-year-old patient with persistent headaches following a major TBI demonstrated a marked improvement in pain and quality of life after ONA treatment when other therapies had failed [24]. Military populations are at high risk for PTH due to their exposure to TBIs. A four-year cohort study involving 64 soldiers treated with ONA (between 155 and 200 units) showed beneficial effects in 41 (64%) of them [25]. A randomized controlled trial in military veterans compared the effects of abobotulinumtoxin A to saline injections over 16 weeks [26]. ONA abobotulinumtoxin A both work by blocking nerve signals to the muscles, but are manufactured differently and they have different potencies and possibly differences in their duration or onset of effect. In this trial, the treatment group experienced a significant reduction in the number of headache days per week and headache intensity, highlighting the potential of botulinum toxin treatments in reducing the burden of PTH. Table 1 summarizes the main characteristics of human studies that evaluate the efficacy of botulinum toxins in the treatment of PTH.

## 3. Anti-CGRP Antibodies

Patients experiencing PTH that resemble migraine can find relief with anti-CGRP mAbs, such as eptinezumab, erenumab, fremanezumab, and galcanezumab. The activation of trigeminovascular nerves by painful stimuli leads to the release of neuropeptides, predominantly CGRP, which facilitates neurovascular and inflammatory changes contributing to head pain [27] The role of CGRP in PTH has been described by animal models of concussion and TBI [28,29,30,31]. In a randomized, double-blind, placebo-controlled study, intravenous infusion of CGRP triggered headache with migraine-like characteristics in patients with persistent PTH who did not have a prior history of migraine [32]. In this crossover study, 30 participants were given an intravenous infusion of either CGRP or placebo for 20 min across two separate experimental days. During the 12-h period following the CGRP infusion, 21 out of 30 participants (70%) experienced a worsening of headache with migraine-like symptoms, compared to only 6 participants (20%) after receiving placebo. These findings were further confirmed in a larger, non-randomized, single-arm, open-label study, which included 60 participants with a documented history of persistent PTH [33]. All participants received an intravenous infusion of CGRP over 20 min, resulting in 43 out of 60 participants (72%) developing migraine-like headaches. A single-center, non-randomized, open-label study suggested that individuals with persistent PTH might benefit from a three-month course of erenumab, a mAb against the CGRP receptor [34]. A total of 89 adults with persistent PTH completed the trial. They received 140 mg of erenumab monthly, administered every four weeks for 12 weeks. Initially, participants reported an average of 24.6 ± 6.1 headache days per month; this number decreased by 1.7 ± 6.9 days by weeks 9 through 12. The average Headache Impact Test-6 (HIT-6) scores, which measure the impact of headaches on daily life, decreased from 61.6 ± 5.2 at baseline to 57.0 ± 8.2. The average number of headache days of moderate-to-severe intensity also dropped from 15.7 days at baseline to 12.9 days by weeks 9 through 12. A case report of a 56-year-old man with a 30-year history of PTH and a case series involving five women with PTH confirmed the beneficial effects of erenumab [35,36]. Conversely, a phase 2 trial found that fremanezumab, a mAb targeting the CGRP peptide, was not more effective than placebo in preventing PTH [37]. A total of 87 eligible participants were randomly assigned to receive either monthly subcutaneous injections of fremanezumab (675 mg) or a placebo over a 12-week double-blind period. Those receiving fremanezumab did not see a greater reduction in moderate-to-severe headache days compared to the placebo group (−3.6 and −5.1 days for fremanezumab and placebo groups, respectively). Secondary outcomes also showed no difference between fremanezumab and placebo groups. The results are available on https://clinicaltrials.gov/ [38]. Currently, CGRP levels in the blood are not considered a reliable biomarker for persistent PTH. In adults with TBI, serum CGRP levels were lower in those with severe TBI compared to those with mild or moderate TBI and healthy controls [39]. Conversely, increased serum CGRP levels were observed in patients with persistent post-concussion symptoms, with levels decreasing over time [40]. Another study compared 100 individuals with persistent PTH to 100 age- and gender-matched healthy controls and found that those with persistent PTH had lower plasma levels of CGRP [41]. Table 2 summarizes the main characteristics of human studies that evaluate the efficacy of anti-CGRP mAbs in the treatment of PTH.

## 4. Considerations on Treatment Adherence and Costs

The evaluation of treatment adherence, persistence, and costs associated with anti-CGRP therapies and onabotulinumtoxin A in patients with PTH remains unknown. Some insights can be drawn from studies conducted in patients with migraine, where both onabotulinumtoxin A and anti-CGRP therapies have been shown to be cost-effective compared to placebo [42]. The annual costs of onabotulinumtoxin A are significantly lower than those of CGRP injections, which can amount to as much as $7000. A real-world 12-month evaluation conducted in patients with chronic migraine indicates that total migraine-related costs were comparable between the onabotulinumtoxin A cohort and the CGRP mAb cohort ($9502 vs. $10,446, respectively) [43]. When dissecting these costs, onabotulinumtoxin A was associated with lower migraine-related pharmacy costs ($1652 vs. $7432), inpatient costs, and lower expenses for both acute ($701 vs. $968) and preventive medication ($343 vs. $453) when compared to CGRP mAbs [43]. The persistence to therapy may be influenced by the route of administration and dosing frequency. Anti-CGRP therapies such as erenumab, fremanezumab, and galcanezumab are administered via monthly subcutaneous injections, with fremanezumab also available as a quarterly option. Eptinezumab, on the other hand, is delivered through quarterly intravenous infusions. Onabotulinumtoxin A requires quarterly intramuscular injections to the face, head, and neck. Adherence to onabotulinumtoxin A has been higher in migraine patients, with a lower rate of discontinuation compared to subcutaneously administered anti-CGRP mAbs [43,44,45]. Similar persistence rates between eptinezumab and onabotulinumtoxin A suggest that the provider-administered nature of these therapies contrasts with the self-administration of subcutaneous anti-CGRP mAbs, which, despite their appealing, may not provide the same level of ongoing patient engagement and adherence [44].

## 5. Conclusions

The treatment of PTH with ONA and anti-CGRP mAbs remains an area with limited clinical evidence. While onabotulinumtoxin A is well-established for chronic migraine, its specific efficacy in PTH is not yet supported by robust clinical trials. The only available trial involving a botulinum toxin in PTH used abobotulinumtoxin A, which, although similar to ONA, underscores the need for studies directly assessing the latter. The number, injection sites, and doses of the toxin varied across the studies reported in Table 1, emphasizing the need to determine the optimal dose and administration method for PTH. Regarding anti-CGRP mAbs, the evidence is even more sparse. A phase 2 trial evaluating fremanezumab, an anti-CGRP monoclonal antibody, failed to demonstrate efficacy in preventing PTH. This negative trial suggests that CGRP inhibition alone may not adequately address the complex pathophysiology of PTH [46,47]. Considering that fremanezumab, targeting the CGRP peptide, failed in a phase 2 trial while erenumab, which targets the CGRP receptor, showed some efficacy (though the study was open-label), the CGRP receptor may play a more relevant role than the peptide itself. Of interest, erenumab’s potent antagonism of the amylin receptor 1 (AMY1) raises questions about its relevance in PTH [48]. Onabotulinumtoxin A, on the other hand, offers the possibility of blocking multiple pain mediators, including substance P and PACAP, in addition to CGRP [49,50]. When considering the broader context of treatment costs and patient adherence, ONA is less expensive than anti-CGRP mAbs and has been associated with higher adherence rates in patients with chronic migraine, likely due to its provider-administered nature, which ensures ongoing patient engagement. In contrast, the self-administration of anti-CGRP mAbs, while convenient, may not offer the same level of sustained adherence, potentially diminishing their long-term effectiveness. While both ONA and anti-CGRP mAbs offer potential avenues for PTH treatment, the current evidence is insufficient to make definitive recommendations. The broader mechanistic action of ONA, coupled with its cost-effectiveness and potential for higher adherence, positions it as a more promising candidate for PTH management. Further studies are needed to confirm its role in this context.

## 6. Methods

We conducted a comprehensive literature search on PubMed to identify relevant studies on ONA, anti-CGRP mAbs, and PTH. The search included articles published up to September 2024. We employed distinct search strategies tailored to each topic area to ensure thorough coverage of the relevant literature. We included articles that focused on the efficacy, safety, and mechanisms of ONA and anti-CGRP mAbs in the treatment of PTH. Both clinical trials and observational studies were considered, provided they were peer-reviewed and available in full text. We focused primarily on human studies, although relevant animal studies were also included. We excluded articles not published in English, as well as reviews and editorials. In addition to the PubMed search, we manually reviewed the reference lists of the retrieved articles to identify any additional relevant studies. Articles of particular interest that were not captured in the initial search but were found in the references were included in the final review. A graphical abstract has been produced using PowerPoint and BioRender (https://biorender.com).

## Figures and Tables

**Table 1 toxins-16-00427-t001:** Main characteristics of human studies that evaluate the efficacy of botulinum toxins in the treatment of post-traumatic headache.

Study	Type of Study	Number of Patients	Diagnosis	Treatment	Outcome
Ferguson et al. [23]	Case series	2 women and 1 man	Persistent post-traumatic headaches, neck pain and involuntary muscle spasm following gunshot wounds to the head, neck and face	90-day injections of onabotulinumtoxin A (from 120 to 155 units)	Each patient reported between 70% and 100% improvement of their headache pain, neck pain and spasm with a significant reduction in the frequency, duration and intensity of their headaches
Lippert-Grüner [24]	Case report	1 woman	Persistent post-traumatic headache	Onabotulinumtoxin A (22 units)	After ten days, the patient was free of symptoms
Yerry et al. [25]	Cohort study	63 men and 1 woman	Persistent post-traumatic headache	90-day injections of onabotulinumtoxin A (from 155 to 200 units)	41 patients (64%) reported feeling better
Zirovich et al. [26]	Randomized, placebo-controlled crossover study	38 men and 2 women	Persistent post-traumatic headache	387.5 units of abobotulinumtoxin A	At the end of the 16 weeks, the number of headaches per week significantly decreased with abobotulinumtoxin A treatment and significantly increased with placebo

**Table 2 toxins-16-00427-t002:** Main characteristics of human studies that evaluate the efficacy of anti-calcitonin gene-related peptide (CGRP) monoclonal antibodies in the treatment of post-traumatic headache.

Study	Type of Study	Number of Patients	Diagnosis	Treatment	Outcome
Ashina et al. [34]	Open-label study	75 women and 25 men	Persistent post-traumatic headache	3 monthly injections of erenumab (140 mg)	Compared to baseline, erenumab resulted in a lower frequency of moderate to severe headache days at week 12
Papenhoff and Dudda [35]	Case report	1 man	Persistent post-traumatic headache	Monthly injections of erenumab (70 mg)	The patient achieved a rapid and stable reduction in his symptoms down to 2–3 headache days per month without adverse events
VanderEnde et al. [36]	Case series	5 women	Persistent post-traumatic headache	Monthly injections of erenumab (70 or 140 mg)	Three of five patients reported a 45–60% decrease in headache intensity since starting treatment with erenumab
Spierings et al. [37,38]	Phase 2 trial	50 women and 37 men	Persistent post-traumatic headache	3 monthly injections of fremanezumab (675 mg)	Fremanezumab treatment did not result in a greater reduction in moderate-to-severe headache days compared to placebo

## Data Availability

No new data were created or analyzed in this study. Data sharing is not applicable to this article.

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
