# Peer review of "Onabotulinumtoxin A for the Treatment of Post-Traumatic Headache: Is It Better than Anti-CGRP Antibodies?"

_toxins, 2024, doi:10.3390/toxins16100427_

Round 1

Reviewer 1 Report

Comments and Suggestions for Authors

Onabotulinumtoxin A for the treatment of post-traumatic headache: is it better than anti-CGRP antibodies?

Overview: A brief review contrasting results from clinical studies on Onabotulinumtoxin A for post-traumatic headache. Based on current studies in the literature, Ona is more effective compared to CGRP antibodies

1.  It would be helpful to discuss other non-neuromuscular junction mechanisms of action of ONA, as it relate to migraine and PTH (ex SNAP25 cleavage etc, and how that modulates the activity of TRP channels).

1.It would be helpful to include the CGRP antibodies in the table (and edit legend to match) to give a comparative overview of the two therapies.

Additionally, numbers per sex would also be helpful in the table.

3. There is a brief discussion on how CGRP is not a reliable marker for persistent PTH, as it is found to be lower in the serum of severe TBI patients. How frequently does persistent headache affect moderate and mild TBI patients, based on literature. With mention of a case series of patients receiving erenumab, which targets the CGRP receptor instead of the peptide, could an increase of receptor as opposed to peptide be in these patients? Perhaps it would be worth a quick discussion point or two.

4. The authors should elaborate more about when the treatment failed in the trials and when it had an effect.

Comments on the Quality of English Language

The quality of the English in the article is sufficiently comprehensible.

Author Response

Comments 1: It would be helpful to discuss other non-neuromuscular junction mechanisms of action of ONA, as it relate to migraine and PTH (ex SNAP25 cleavage etc, and how that modulates the activity of TRP channels).
Response 1: Thanks for pointing this out. We have clarified the mechanism of action of onabotulinumtoxin A (ONA) in relation to SNAP25 cleavage (page 2, lines 86-90).

Comments 2: It would be helpful to include the CGRP antibodies in the table (and edit legend to match) to give a comparative overview of the two therapies.
Response 2: Thanks for your comment. Table 2 already includes an overview of the results of CGRP antibodies in the context of post-traumatic headache (PTH). As this review does not aim to provide a statistical comparison between ONA and CGRP antibodies, we believe that the two tables are sufficient to give readers a comparative overview of the current findings regarding ONA and CGRP antibodies for PTH treatment.

Comments 3: Additionally, numbers per sex would also be helpful in the table.
Response 3: Thanks for pointing this out. We have updated Table 1 and Table 2 accordingly.

Comments 4: There is a brief discussion on how CGRP is not a reliable marker for persistent PTH, as it is found to be lower in the serum of severe TBI patients. How frequently does persistent headache affect moderate and mild TBI patients, based on literature. With mention of a case series of patients receiving erenumab, which targets the CGRP receptor instead of the peptide, could an increase of receptor as opposed to peptide be in these patients? Perhaps it would be worth a quick discussion point or two.
Response 4: Thanks for pointing this out. We have added some considerations about the complex pathophysiology of PTH, including the relevance of the CGRP peptide, the CGRP receptor and the AMY1 receptor at page 5 (lines 199-204).

Comments 5: The authors should elaborate more about when the treatment failed in the trials and when it had an effect.
Response 5: Thanks for your comment. We reported outcomes in Table 1 and Table 2. In addition, we discussed outcomes in the final remarks. Regarding ONA, two positive studies have been reported (a cohort study and a randomized, placebo-controlled trial). Considering CGRP antibodies, a positive open-label study and a negative phase 2 trial have been described. Given the narrative focus of this review, we believe it is not necessary to provide further elaboration on this topic at this stage. Future systematic reviews with more comprehensive studies may explore it in greater detail.

Reviewer 2 Report

Comments and Suggestions for Authors

I suggest to add a paragraph regarding the mecchanisms of action of botulinum in PTH. 
The grammar is mostly correct. Moreover, there is the need to discuss the route of administration of botulinum and the dose used in different studies: there were differences?

in the methods, how many articles were examined? How many were excluded? This is not clear.

Author Response

Comments 1: I suggest to add a paragraph regarding the mechanisms of action of botulinum in PTH. 
Response 1: Thanks for your comment. We have clarified the mechanisms of action of onabotulinumtoxin A (ONA). As the focus of this review is not to provide an in-depth analysis of the mechanisms of action of botulinum toxin in PTH, we have decided not to include a dedicated paragraph on this topic.

Comments 2: Moreover, there is the need to discuss the route of administration of botulinum and the dose used in different studies: there were differences?
Response 2: Thanks for pointing this out. Yes, there were differences (as displayed from Table 1). We have added this point at page 5 (lines 193-195).

Comments 3: In the methods, how many articles were examined? How many were excluded? This is not clear.
Response 3: Thanks for pointing this out. As this review has a narrative scope, we did not focus on quantifying the number of articles examined or excluded, as the aim was to explore this topic for the first time. However, we performed independent searches on PubMed with different search strategies to ensure the robustness of the data included in the review.